# Molecular Profile of Variants Potentially Associated with Severe Forms of COVID-19 in Amazonian Indigenous Populations

**DOI:** 10.3390/v16030359

**Published:** 2024-02-26

**Authors:** Rita de Cássia Calderaro Coelho, Carlliane Lima e Lins Pinto Martins, Lucas Favacho Pastana, Juliana Carla Gomes Rodrigues, Kaio Evandro Cardoso Aguiar, Amanda de Nazaré Cohen-Paes, Laura Patrícia Albarello Gellen, Francisco Cezar Aquino de Moraes, Maria Clara Leite Calderaro, Letícia Almeida de Assunção, Natasha Monte, Esdras Edgar Batista Pereira, André Maurício Ribeiro-dos-Santos, Ândrea Ribeiro-do-Santos, Rommel Mario Rodriguez Burbano, Sandro José de Souza, João Farias Guerreiro, Paulo Pimentel de Assumpção, Sidney Emanuel Batista dos Santos, Marianne Rodrigues Fernandes, Ney Pereira Carneiro dos Santos

**Affiliations:** 1Oncology Research Center, Federal University of Pará, Belém 66073-005, PA, Brazil; rccalderarocoelho@gmail.com (R.d.C.C.C.); carllianepmartins@hotmail.com (C.L.e.L.P.M.); lucas.pastana@ics.ufpa.br (L.F.P.); julianarodrigues@gmail.com (J.C.G.R.); kaio.evandro@hotmail.com (K.E.C.A.); acohencastro@gmail.com (A.d.N.C.-P.); laura.patricia.agellen@hotmail.com (L.P.A.G.); francisco.cezar2205@gmail.com (F.C.A.d.M.); maria.calderaro@ics.ufpa.br (M.C.L.C.); leticiaalmeidaenf96@gmail.com (L.A.d.A.); ntshmonte@gmail.com (N.M.); edgarpereira@ufpa.br (E.E.B.P.); rommelburbano@gmail.com (R.M.R.B.); assumpcaopp@gmail.com (P.P.d.A.); sidneysantos@ufpa.br (S.E.B.d.S.); fernandesmr@yahoo.com.br (M.R.F.); 2Laboratory of Human and Medical Genetics, Institute of Biological Science, Federal University of Pará, Belém 66077-830, PA, Brazil; andremrsantos@gmail.com (A.M.R.-d.-S.); akelyufpa@gmail.com (Â.R.-d.-S.); joao.guerreiro53@gmail.com (J.F.G.); 3Ophir Loyola Hospital, Pará State Department of Health, Belém 66063-240, PA, Brazil; 4Brain Institute, Federal University of Rio Grande do Norte, Natal 59078-970, RN, Brazil; sandro@neuro.ufrn.br

**Keywords:** COVID-19, SARS-CoV-2, coronavirus, indigenous population, SNP

## Abstract

Coronavirus disease 2019 (COVID-19) is an infection caused by SARS-CoV-2. Genome-wide association studies (GWASs) have suggested a strong association of genetic factors with the severity of the disease. However, many of these studies have been completed in European populations, and little is known about the genetic variability of indigenous peoples’ underlying infection by SARS-CoV-2. The objective of the study is to investigate genetic variants present in the genes *AQP3*, *ARHGAP27*, *ELF5L*, *IFNAR2*, *LIMD1*, *OAS1* and *UPK1A,* selected due to their association with the severity of COVID-19, in a sample of indigenous people from the Brazilian Amazon in order to describe potential new and already studied variants. We performed the complete sequencing of the exome of 64 healthy indigenous people from the Brazilian Amazon. The allele frequency data of the population were compared with data from other continental populations. A total of 66 variants present in the seven genes studied were identified, including a variant with a high impact on the *ARHGAP27* gene (rs201721078) and three new variants located in the Amazon Indigenous populations (INDG) present in the *AQP3*, *IFNAR2* and *LIMD1* genes, with low, moderate and modifier impact, respectively.

## 1. Introduction

In December 2019, in Wuhan, China, the first cases of severe acute respiratory infection (SARS-CoV-2) by coronavirus (COVID-2) were diagnosed [1,2]. In a short time, the cases of infection reached large proportions, spreading to several regions and reaching several countries [3,4]. According to data from the World Health Organization (WHO), on June 7th, 2023, 767,750,853 cases of SARS-CoV-2 infection and 6,941,095 COVID-19 deaths were confirmed worldwide [5]. Therefore, even with the improvement of the reality of the pandemic, due to the administration of vaccine doses, this disease still represents a global public health problem [6].

There is great clinical diversity among patients with COVID-19. The disease can manifest in different ways, varying from asymptomatic to mild and severe forms, and can lead to death [3,4]. Clinical studies relate the heterogeneity of the disease with the genetic influence on the individual response to infection. The association of genetic factors with the severity and clinical evolution of COVID-19 has been investigated little. However, genome-wide association studies (GWASs) have been developed to understand better the relationship of genes associated with the severity of this disease [7,8,9], enabling the development of more specialized therapies for the risk group [7].

The association of genetic factors with the severity of COVID-19 has been addressed in different studies [7,8,9]. Another study developed by our research group [10] also investigated the interaction of genes (*SLC6A20*, *LZTFL1*, *CCR9*, *FYCO1*, *CXCR6*, *XR1* and *ABO*) with the most severe forms of the disease and demonstrated a strong relationship of locus 3p21.31 with the severity of SARS-CoV-2 infection [7,8].

Seven new genes (*AQP3*, *ARHGAP27*, *ELF5*, *IFNAR2*, *LIMD1*, *OAS1* and *UPK1A*) were related to the severity of COVID-19 in populations of European origin. The difference in disease severity between genders and the association of androgenic hormones with the severity of SARS-CoV-2 infection were also addressed in the study [9].

Therefore, the objective of our study is to investigate genetic variants in the genes *AQP3*, *ARHGAP27*, *ELF5*, *IFNAR2*, *LIMD1*, *OAS1* and *UPK1A,* which were selected due to the association of these genes with the severity of COVID-19, in a sample of Amazonian indigenous peoples.

## 2. Materials and Methods

### 2.1. Consent and Ethics

This study was approved by the National Research Ethics Committee (CONEP) and the Research Ethics Committee of the Center for Tropical Medicine with the opinion 20654313.6.0000.5172 and CAAE 33934420.0.0000.5634. The representatives of the groups participating in the study were informed about the stages of the study and signed the Free and Informed Consent Form (TCLE). Their materials were collected according to the Declaration of Helsinki.

### 2.2. Study Population

This study was carried out through the blood collection from 64 healthy indigenous people from the Brazilian Amazon region, belonging to the original groups: Asurini do Tocantins, Asurini do Xingu, Araweté, Arara, Juruna, Awa-Guajá, Kayapó/Xikrin, Munduruku, Karipuna, Phurere, Wajãpi and Zo’é. The collection of blood material was carried out before the pandemic period. The study participants were healthy and did not have COVID-19.

Information from markers indicative of ancestry (AIMs) was obtained to confirm the ancestry and the mixture between continental populations (European, African and Asian) in three multiplex PCR reactions [11,12,13]. Electrophoresis was used to analyze the amplicons in the sequencer ABI Prism 3130 and GeneMapper ID v.3.2. In addition, the proportions of the individuals were analyzed in the STRUCTURE v.2.33 software. The allele frequency data of the INDG population were obtained by the allele count and compared with data from 5 other continental populations (AFR, AMR, EAS, EUR and SAS) found in the Project 1000 Genomes database (http://www.1000genomes.org; accessed on March 30th, 2022).

### 2.3. DNA Extraction and Preparation of the Exome Library

The blood collection of each participant in this study was carried out using 5 mL tubes. Subsequently, this material was extracted with the Roche Applied Science DNA extraction kit (Roche, Penzberg, Germany), according to the manufacturer’s instructions. The samples were quantified in NanoDrop1000 to verify the integrity of the genetic material. The exome library was prepared with the help of the Nextera Rapid Capture Exome kit (Illumina, San Diego, CA, USA) and the SureSelect Human All Exon V6 kit (Agilent, Santa Clara, CA, USA). The sequencing step was developed on the NextSeq 500® (Illumina®, San Diego, CA, USA) using the NextSeq 500 v2 300 high production cycle kit (Illumina®, San Diego, CA, USA).

### 2.4. Bioinformatics Analysis

The quality of the FASTQ reads was analyzed (FastQC v.0.11 http://www.bioinformatics.babraham.ac.uk/projects/fastqc/), and the samples were filtered to eliminate low-quality readings (fastx_tools v.0.13). Subsequently, the sequences that showed good quality were mapped and aligned according to a reference genome (GRCh38) using the BWA v.0.7 software. The variants were identified in GATK v. 3.2 and noted in the Viewer of Variants software (ViVa®, Federal University of Rio Grande do Norte, Natal, Brazil). Other databases were also used: SnpEff v.4.3. T, Ensembl Variant Effect Predictor (Ensembl version 99) and ClinVar (v.2018–10). The SIFT (v.6.2.1), PolyPhen-2 (v.2.2), LRT (November 2009), Mutation Assessor (v.3.0), Mutation Taster (v.2.0), FATHMM (v.2.3), PROVEAN (v.1.1.3), MetaSVM (v.1.0), M-CAP (v.1.4) and FATHMM-MKL. Bioinformatics analyses were performed as described in Ribeiro-Dos-Santos et al. [14] and Rodrigues et al. [13].

### 2.5. Statistical Analyses

In the statistical analysis, two tests were used: the first was Fisher’s exact test, to differentiate the frequencies between the populations of the world. The results obtained were considered statistically significant when *p* ≤ 0.05. Subsequently, the Wright fixation index (FST) was used to estimate population differentiation. The statistical analyses of this study were developed in the software Arlequin v.3.518 and R Studio to develop the graphic data.

### 2.6. Selection of Variants

Seven genes were used in this study (*AQP3*, *ARHGAP27*, *ELF5*, *IFNAR2*, *LIMD1*, *OAS1* and *UPK1A*), selected based on the study by Cruz et al. [9]. The variants were selected based on three main criteria: (a) at least 10 coverage readings (fastx_tools v.0.13-http://hannonlab.cshl.edu/fastx_toolkit/; accessed on January 20th, 2022); (b) an allelic frequency described in the continental populations of the 1000 Genomes Project Consortium [15]; (c) variants must have the modifier impact, moderate or high, as classified by SnpEff [16]. The SnpEff is a type of classification that evaluates the effect of variants, genes and genetic changes.

## 3. Results

In our study, we identified 66 variants distributed in the seven genes (*AQP3*, *ARHGAP27*, *ELF5*, *IFNAR2*, *LIMD1*, *OAS1* and *UPK1A*), of these, 7 are present in the *AQP3* gene, 14 in *ARHGAP27*, 6 in *ELF5*, 14 in the *IFNAR2* gene, 11 in the *LIMD1* gene, 9 in *OAS1* and 5 in the *UPK1A* gene. Seventeen variants were excluded due to their low coverage (Appendix A). After going through these quality criteria and impact prediction, we identified 45 variants that were included in the study (Table 1) and three exclusive variants of the indigenous population (Table 2) and compared them with the other world populations described in the 1000 Genomes database (African (AFR), American population (AMR), East Asian (EAS), European (EUR) and South Asian (SAS)).

The 45 variants described in Table 1 were characterized by information such as the chromosomal position, SNP ID, nucleotide alteration and classification by SNPeff, excluding the low-impact ones (except the new variants). A high-impact variant was identified in the *ARHGAP27* gene (rs201721078) at position 45404053 with a frequency of 21.8% in the INDG population, being rare in the rest of the world.

Thirty-six modifier impact variants were also identified in the seven genes studied: three in *AQP3* (rs2231235, rs2231231 and rs12555686), eight *ARHGAP27* (rs7222444, rs62064597, rs115993362, rs35327136, rs7213200, rs142163608, rs184103721 and rs35389313), five in *ELF5* (rs2231825, rs2231821, rs556840829, rs28395819 and rs737254), eight in *IFNAR2* (rs1131668, rs9984273, rs2252639, rs2834158, rs17860118, rs3216172 rs397789038, rs12482556, rs56197608 and a new variant), three in *LIMD1* (rs2742409, rs2578679 and a new variant), six in *OAS1* (rs7968145, rs1131476, rs10774671, rs7967461, rs11352835 and rs1051042) and three in *UPK1A* (rs747589460, rs75289222 and rs2285421).

Of the variants found in this study, rs12949256 (*ARHGAP27*), rs2231235, rs12555686 (*AQP3*) and rs56197608 (*IFNAR2*) were not present in the INDG population. We also identified rs1051393 (78.9%), rs1131668 (79.5%) and rs12482556 (71.4%), which were present at high allele frequencies in the *IFNAR2* gene in the indigenous populations. In our study, in the *OAS1* gene, rs10774671 (83.3%), rs2660 (100%), rs7967461 (100%), rs11352835 (100%) and rs1051042 (100%), also showed high frequencies in indigenous people.

We also identified three new variants in the Amazonian indigenous people (Table 2). The first was in the *AQP3* gene at position 33442882 with a low impact and allele frequency of 8.3%. The second variant was identified in the *IFNAR2* gene at position 33262799 with a moderate impact and a frequency of 8.3%, and the third was found in the *LIMD1* gene at position 45676789 with a modifier impact and frequency of 6.6%.

In addition, 10 variants of moderate impact were also found in five of the seven genes studied: three in the *ARHGAP27* gene (rs2959953, rs12949256 and rs117139057), three in the *IFNAR2* gene (rs1051393, rs2229207 and the presence of a new variant), one in LIMD1 (rs267237), two in *OAS1* (rs1131454 and rs2660), and one in *UPK1A* (rs2267586).

Among the variants identified, 17 showed significant differences when compared to other world populations (AFR, AMR, SAS, EAS and EUR), even in East Asians, who have greater genetic similarity with indigenous people. Two variants were identified in the *ARHGAP27* gene (rs201721078 and rs2959953), three were present in *AQP3* (rs2228332, rs591810 and rs2231231), one in the *UPK1A* gene (rs2285421), four in *OAS1* (rs2660, rs7967461, rs11352835 and rs1051042) and seven were found in the *IFNAR2* gene (rs1051393, rs2229207, rs1131668, rs9984273, rs2834158, rs149186597, rs79402470, rs3216172 and rs397789038) (Table 3). The frequencies of the other variants did not show significant differences between the INDG population and the other continental populations (Appendix A).

The genetic differences between the populations (Figure 1) of the study were analyzed using the multidimensional scale graph (MDS), based on the Fisher fixation test of the genetic variants. The MDS identified greater similarity between the INDG and AMR populations, mainly due to the influence of indigenous peoples on the AMR populations. A difference was also identified between EUR, AFR, SAS and the EAS population that showed a greater difference when compared to the other populations.

## 4. Discussion

GWASs with genetic variants associated with the severity of COVID-19 have been developed in several world populations [7,8,9]. However, there is a gap in the information about the investigations of genetic variants related to COVID-19 in indigenous people. Therefore, studies that seek to relate the influence of genetics on the individual response to the disease within these population groups are very important.

These population groups have been suffering from cases of diseases such as measles, flu and tuberculosis [10,17]. Factors such as the unique genetic profile, as well as the presence of rare or little-known mutations in indigenous people, can contribute to the incidence of infectious diseases in these populations [17]. Geographical isolation and the existence of consanguineous marriages can also favor the differentiation of the allele frequency in these populations, when compared to other world populations [18,19].

A recent study developed by our research group investigated genetic variants present in the genes *SLC6A20*, *LZTFL1*, *CCR9*, *FYCO1*, *CXCR6*, *XR1* and *ABO* involved with the severity of COVID-19 in indigenous people [10]. The results of the study demonstrated that the variants identified by Ellinghaus et al. [7] were not found in indigenous people. In addition, such data suggested a low genetic variability in these populations, evidencing their unique genetic profile.

Another study developed in 34 Spanish hospitals with 11,939 positive cases of COVID-19 identified the relationship of the *AQP3*, *ARHGAP27*, *ELF5*, *IFNAR2*, *LIMD1*, *OAS1* and *UPK1A* genes with the severity of the disease. In addition, this study suggested the difference in the severity of COVID-19 between the sexes. Due to the greater propensity to develop a more critical picture of SARS-CoV-2 infection in males, the results of this investigation also indicated the relationship of androgenic hormones with the severity of the disease [9]. Such data suggest that the genes *AQP3*, *ARHGAP27*, *ELF5*, *IFNAR2*, *LIMD1*, *OAS1* and *UPK1A*, related to sexual differences in Spaniards in the development of more severe forms of COVID-19, may also be involved with androgenic hormonal pathways.

We also identified three new variants that may be potentially related to the severity of COVID-19 among indigenous people of the Brazilian Amazon. The first genetic variant was identified in the *AQP3* gene. Some studies have already demonstrated the relationship of *AQPs* with diseases such as cancer, metabolic syndrome and epilepsy [20,21,22]. Despite the limitation of information on *AQPs* and infectious diseases, the cellular inability to maintain the movement of fluids from the human body can alter homeostasis [23]. Therefore, *AQPs* may be extremely necessary for the control of homeostasis in cases of infectious diseases such as COVID-19 [24]. Recently, in a GWAS study, 49 variants were associated with the most severe forms of COVID-19, showing that the *AQP3* gene had an intense relationship with the most critical picture of infection, a fact that corroborates other studies [9,25].

A second new variant was also identified in *IFNAR2* and the variants rs1051393, rs1131668 and rs12482556 showed high allelic frequencies in the indigenous population. Recent studies have reported the association of the *IFNAR2* gene with the most severe forms of COVID-19, as well as the relationship of generic variants with more critical cases of the disease [26,27,28]. The *OAS1* gene was also identified in the indigenous populations studied and five genetic variants (rs10774671, rs2660, rs7967461, rs11352835 and rs1051042) showed a high allele frequency within this population group. Recent studies point to the association of the *OAS1* gene, as well as rs10774671, with the severity of COVID-19 [29].

The third and last new variant was found in *LIMD1*, a gene with reports of involvement in cellular processes and the progression of diseases such as cancer [30,31,32]. However, its relationship with COVID-19 has not yet been well elucidated and future studies are needed to prove the association of this gene with the severity of the disease.

Finally, a high-impact variant never before associated with COVID-19 was also identified in the *ARHGAP27* gene (rs201721078). With this, the results of this study can contribute with important information to assess the risk of developing more severe forms of COVID-19 in indigenous populations of the Amazon.

## 5. Conclusions

This study investigated genes potentially related to the severity of COVID-19 in an Amazonian indigenous. The results found in this study suggest the urgency of more effective research that proves the impact of these new and high-impact variants in patients with SARS-CoV-2 infection in the indigenous populations of the Amazon, aiming to elucidate the biological role of these variants in the severity of COVID-19 in indigenous people and contributing to the development of personalized medicine that respects the particularities of the studied population.

## Figures and Tables

**Figure 1 viruses-16-00359-f001:**
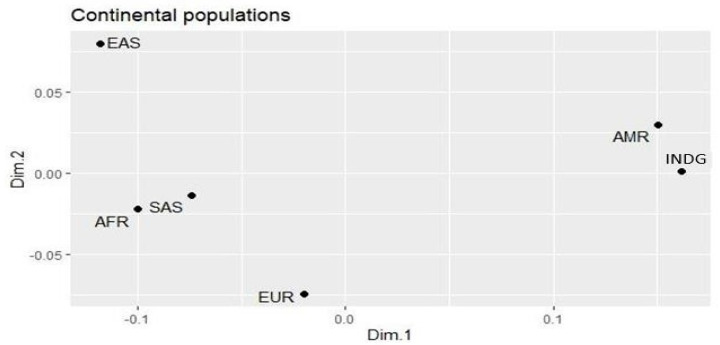
Differences in the allelic frequencies of the variants studied in the indigenous population and the continental populations, plotted in multidimensional scaling plot (MDS).

**Table 1 viruses-16-00359-t001:** Description of the variants according to the high, moderate and modifier impact present in the *AQP3*, *ARHGAP27*, *ELF5*, *IFNAR2*, *LIMD1*, *OAS1* and *UPK1A* genes.

Gene	Position	SNP ID	Ref ^a^	Var ^b^	Impact Predicted by SNPeff	Variant Allele Frequence					
						INDG	AFR	AMR	EAS	EUR	SAS
*ARHGAP27*	45404053	rs201721078	G	A	HIGH	0.218	0.0009	0.0421	-	0.0001	-
*ARHGAP27*	45429642	rs2959953	G	C	MODERATE	0.375	0.6953	0.6537	0.520	0.717	0.673
*ARHGAP27*	45429931	rs12949256	C	T	MODERATE	0	-	-	-	0.5	0.166
*ARHGAP27*	45395459	rs117139057	G	C	MODERATE	0.125	0.089	0.2028	0.2768	0.1791	0.1189
*ARHGAP27*	45430283	rs7222444	A	G	MODIFIER	0.009	0.102	0.158	0.0005	0.243	0.071
*ARHGAP27*	45395941	rs62064597	C	G	MODIFIER	0.083	0.0424	0.132	0.0008	0.2141	0.073
*ARHGAP27*	45429583	rs115993362	C	G	MODIFIER	0.083	0.147	0.0062	-	0.0002	0.0006
*ARHGAP27*	45410208	rs35327136	C	A	MODIFIER	0.015	0.1308	0.1778	-	0.2233	0.0682
*ARHGAP27*	45396860	rs7213200	G	A	MODIFIER	0.016	-	-	-	-	-
*ARHGAP27*	45410198	rs142163608	C	T	MODIFIER	0.083	0.0162	-	-	-	-
*ARHGAP27*	45404234	rs184103721	C	G	MODIFIER	0.045	0.0004	0.048	0.010	0.0002	0.0004
*ARHGAP27*	45410058	rs35389313	T	C	MODIFIER	0.013	-	-	-	-	-
*AQP3*	33442274	rs2231235	G	A	MODIFIER	0	0.012	0.109	0.063	0.069	0.085
*AQP3*	33442988	rs2231231	A	C	MODIFIER	0.937	0.775	0.7272	0.721	0.639	0.586
*AQP3*	33447549	rs12555686	G	T	MODIFIER	0	0.0221	0.255	0.175	0.095	0.095
*IFNAR2*	33241950	rs1051393	T	G	MODERATE	0.789	0.193	0.487	0.583	0.336	0.476
*IFNAR2*	33241945	rs2229207	T	C	MODERATE	0.343	0.079	0.166	0.178	0.084	0.130
*IFNAR2*	33262573	rs1131668	G	A	MODIFIER	0.795	0.376	0.453	0.583	0.365	0.476
*IFNAR2*	33262760	rs9984273	C	T	MODIFIER	1	0.594	0.848	0.918	0.687	0.767
*IFNAR2*	33245424	rs2252639	A	T	MODIFIER	0.285	1	1	1	1	1
*IFNAR2*	33244908	rs2834158	T	C	MODIFIER	0.333	0.807	0.512	0.416	0.663	0.524
*IFNAR2*	33230489	rs17860118	G	T	MODIFIER	0.187	0.076	0.118	0.139	0.0864	0.120
*IFNAR2*	33262748	rs3216172 rs397789038	G	GT	MODIFIER	0.181	0.366	0.467	0.594	0.33	0.484
*IFNAR2*	33230629	rs12482556	T	C	MODIFIER	0.714	-	-	-	-	-
*IFNAR2*	33252224	rs56197608	C	T	MODIFIER	0	-	0.003	0.021	0.0002	0.0001
*ELF5*	34489994	rs2231825	C	A	MODIFIER	0.046	0.116	0.186	0.027	0.429	0.301
*ELF5*	34505605	rs2231821	A	G	MODIFIER	0.043	0.059	0.155	0.026	0.316	0.265
*ELF5*	34493816	rs556840829	A	G	MODIFIER	0.05	-	-	-	-	-
*ELF5*	34511455	rs28395819	A	C	MODIFIER	0.083	-	-	-	-	-
*ELF5*	34493394	rs737254	T	G	MODIFIER	0.040	0.286	0.327	0.051	0.467	0.339
*LIMD1*	45595761	rs267237	C	T	MODERATE	0.937	0.584	0.721	0.938	0.562	0.675
*LIMD1*	45674275	rs2742409	G	A	MODIFIER	0.083	-	-	-	-	-
*LIMD1*	45674196	rs2578679	G	A	MODIFIER	0.5	-	-	-	-	-
*OAS1*	112919637	rs2660	G	A	MODERATE	1	0.934	0.840	0.791	0.679	0.707
*OAS1*	112911065	rs1131454	G	A	MODERATE	0.645	0.228	0.703	0.614	0.575	0.572
*OAS1*	112931839	rs7968145	C	T	MODIFIER	1	0.989	0.955	1	0.924	0.955
*OAS1*	112919404	rs1131476	G	A	MODIFIER	1	0.934	0.8318	0.784	0.659	0.698
*OAS1*	112919388	rs10774671	G	A	MODIFIER	0.833	0.424	0.799	0.783	0.654	0.701
*OAS1*	112931954	rs7967461	G	C	MODIFIER	1	0.917	0.739	0.76	0.575	0.692
*OAS1*	112931910	rs11352835	TA	T	MODIFIER	1	0.916	0.738	0.769	0.59	0.691
*OAS1*	-	rs1051042	G	C	MODIFIER	1	0.934	0.831	0.784	0.657	0.693
*UPK1A*	35668466	rs2267586	T	G	MODERATE	0.179	0.0715	0.1157	0.2241	0.1007	0.1026
*UPK1A*	35676195	rs747589460	CTTT	C	MODIFIER	0.033	0.0227	0.037	0.0417	0.0587	0.0811
*UPK1A*	35673206	rs75289222	G	T	MODIFIER	0.035	0.0653	0.115	0.1029	0.1016	0.0845
*UPK1A*	35678012	rs2285421	T	C	MODIFIER	0.3	0.8012	0.5853	0.6216	0.5365	0.7368

^a^ Reference Allele; ^b^ Variant Allele; (-) no annotation; INDG: indigenous population; AFR: African population; AMR: American population; EAS: East Asian population; EUR: European population; SAS: South Asian population.

**Table 2 viruses-16-00359-t002:** Description of new variants.

Gene	Position	SNP ID	Ref ^a^	Var ^b^	Allele Frequence	Impact Predicted by SNPeff
*AQP3*	33442882	*	G	A	8.3%	LOW
*IFNAR2*	33262799	*	C	A	8.3%	MODERATE
*LIMD1*	45676789	*	G	T	6.6%	MODIFIER

^a^ Reference Allele; ^b^ Variant Allele; * variants without described SNP.

**Table 3 viruses-16-00359-t003:** Significant difference in allele frequencies of the indigenous population and in the world population (AFR, AMR, EUR, EAS and SAS).

Gene	SNP Id	INDG vs. AFR *	INDG vs. AMR *	INDG vs. EAS *	INDG vs. EUR *	INDG vs. SAS *
*ARHGAP27*	rs201721078	**5.129 × 10^−15^**	**1.542 × 10^−5^**	NA	**1.381 × 10^−14^**	NA
*AQP3*	rs2228332	**0.00018**	**3.693 × 10^−5^**	**5.348 × 10^−5^**	**4.293 × 10^−9^**	**9.082 × 10^−10^**
*AQP3*	rs591810	**3.963 × 10^−5^**	**0.00075**	**1.031 × 10^−5^**	**3.214 × 10^−7^**	**1.087 × 10^−8^**
*AQP3*	rs2231231	**0.00117**	**0.00010**	**5.442 × 10^−5^**	**1.620 × 10^−7^**	**2.264 × 10^−9^**
*UPK1A*	rs2285421	**3.016 × 10^−16^**	**3.180 × 10^−5^**	**1.341 × 10^−6^**	**0.00030**	**1.562 × 10^−11^**
*OAS1*	rs2660	**0.02583**	**7.618 × 10^−5^**	**1.374 × 10^−6^**	**1.525 × 10^−10^**	**2.482 × 10^−9^**
*OAS1*	rs7967461	**0.01055**	**3.650 × 10^−8^**	**1.260 × 10^−7^**	**1.118 × 10^−14^**	**6.043 × 10^−10^**
*OAS1*	rs11352835	**0.01087**	**3.432 × 10^−8^**	**2.366 × 10^−7^**	**3.623 × 10^−14^**	**6.043 × 10^−10^**
*OAS1*	rs1051042	**0.02583**	**4.458 × 10^−5^**	**7.665 × 10^−7^**	**3.220 × 10^−11^**	**5.950 × 10^−10^**
*ARHGAP27*	rs2959953	**6.052 × 10^−7^**	**4.116 × 10^−5^**	**0.03356**	**1.238 × 10^−7^**	**6.31360 × 10^−6^**
*IFNAR2*	rs1051393	**1.824 × 10^−21^**	**1.053 × 10^−5^**	**0.00258**	**1.215 × 10^−11^**	**3.942 × 10^−6^**
*IFNAR2*	rs2229207	**2.456 × 10^−8^**	**0.00186**	**0.003**	**1.035 × 10^−7^**	**6.205 × 10^−5^**
*IFNAR2*	rs1131668	**1.229 × 10^−10^**	**2.771 × 10^−7^**	**0.00098**	**7.147 × 10^−11^**	**9.955 × 10^−7^**
*IFNAR2*	rs9984273	**3.001 × 10^−14^**	**0.00013**	**0.00938**	**3.134 × 10^−10^**	**2.268 × 10^−7^**
*IFNAR2*	rs2834158	**5.32 × 10^−15^**	**0.00664**	**0.22382**	**3.818 × 10^−7^**	**0.00342**
*IFNAR2*	rs149186597 rs79402470	**8.004 × 10^−5^**	**0.00602**	**1.575 × 10^−5^**	**0.04188**	**0.04602**
*IFNAR2*	rs3216172 rs397789038	**0.00380**	**2.699 × 10^−5^**	**4.368 × 10^−10^**	**0.02185**	**3.997 × 10^−6^**

INDG: indigenous population; AFR: African population; AMR: American population; EAS: East Asian population; EUR: European population; SAS: South Asian population; * *p*-value defined by Fisher’s exact test. Bold characters indicate a significant difference (*p*-value * < 0.05).

## Data Availability

The data obtained from the public domain are available at gnomAD (http://broadinstitute.org; accessed on April 27th, 2018), and the sequencing data of the Amazonian Amerindian populations are available at the ENA database under the accession number PRJEB35045.

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
