# Peer review of "Molecular Profile of Variants Potentially Associated with Severe Forms of COVID-19 in Amazonian Indigenous Populations"

_viruses, 2024, doi:10.3390/v16030359_

Round 1

Reviewer 1 Report

Comments and Suggestions for Authors

1. Methods:The authors should better explain /describe how they assessed the genetic polymorphisms (sections 2.4-6)

2. Did the authors chick the expression of OAS1, IFNAR2,  and their relationship with the polymorphisms? If not , why?

3. Was the severity of COVID-19 illness linked to other co-morbidities?( for example tuberculosis)

Reviewer 2 Report

Comments and Suggestions for Authors

In this work, using exome sequencing, the authors estimate SNP’s frequencies in an indigenous population (64 individuals) at seven genes (AQP3, ARHGAP27, ELF5L, IFNAR2, LIMD1, OAS1 and UPK1A), previously shown to harbour SNP´s associated with COVID-19 severity in a European sample. Significant differences in SNP frequencies are found at these genes in the indigenous population and in American, European, African and Asian populations. Therefore, the authors suggest that this might be the reason for the high mortality due to COVID-19 observed in indigenous populations. Nevertheless, this hypothesis is not directly tested in this work, since the authors only performed the exome sequencing of healthy indigenous people. Intriguingly, the authors do not state which of the SNPs found in the indigenous populations were actually shown to be associated with COVID-19 severity in the European sample, thus this hypothesis is not tested indirectly either. Looking at SNP frequencies, it can be seen that some SNPs do show large differences in frequency between populations while others only show small frequency differences (for instance, less than 5%).  It is thus important to say which SNP’s (those showing large differences or those showing small differences in frequency between populations) have been shown to be to be associated with COVID-19 severity in the European sample, since a small difference in frequency in different populations is unlikely to explain the mortality difference in the indigenous and the remaining populations. Assuming, as it seems to be the case, that all variants at a given gene are associated with COVID-19 severity, just because a few were found to be associated with COVID-19 severity in another unrelated population is a dangerous assumption. Therefore, the authors are simply describing SNP frequencies at seven genes based on a relatively small sample size.

Specific major comments:

1)      The following sentence in the abstract is problematic:

“The objective of the study is to investigate genetic variants present in the AQP3, ARHGAP27, ELF5L, IFNAR2, LIMD1, OAS1 and UPK1A genes selected due to the association with the severity of COVID-19 in a sample of indigenous people from the Brazilian Amazon”

If a comma is placed after COVID-19, it is easier to understand, but without it, it seems that the association was found in a sample of indigenous people, which is not the case, but it may be better to rewrite it.

2)      Also in the abstract, please define the acronym INDG, the first time the sample of indigenous people from the Brazilian Amazon is mentioned.

3)      No references are given for the software applications used and no hyperlinks are given for the databases used. Please provide them.

4)      Line 114 – Please state how low quality sequences were eliminated

5)      Line 129 –  Saying “…was used to verify the interpopulation variability of genetic variants” seems to be a complex way of saying “.. was used to estimate population differentiation”

6)      Lines 172-172 – Again saying “…did not have an allelic frequency described in the INDG population” seems to be a complex way of saying: “… were not present in the INDG population”

Minor comments:

Line 67, change “demonstrate” to “demonstrated”

Round 2

Reviewer 1 Report

Comments and Suggestions for Authors

The paper should be considered for publication